# Progress in Understanding Oxidative Stress, Aging, and Aging-Related Diseases

**DOI:** 10.3390/antiox13040394

**Published:** 2024-03-25

**Authors:** Jianying Yang, Juyue Luo, Xutong Tian, Yaping Zhao, Yumeng Li, Xin Wu

**Affiliations:** 1School of Medical Technology and Engineering, Henan University of Science and Technology, Luoyang 471023, China; yyjianying2001@haust.edu.cn (J.Y.); luojy24@tib.cas.cn (J.L.); tianxt@tib.cas.cn (X.T.); 2Tianjin Institute of Industrial Biotechnology, Chinese Academy of Sciences, National Center of Technology Innovation for Synthetic Biology, Tianjin 300308, China; zhaoyp24@tib.cas.cn

**Keywords:** antioxidants, reactive oxygen species, aging, aging-related diseases

## Abstract

Under normal physiological conditions, reactive oxygen species (ROS) are produced through redox reactions as byproducts of respiratory and metabolic activities. However, due to various endogenous and exogenous factors, the body may produce excessive ROS, which leads to oxidative stress (OS). Numerous studies have shown that OS causes a variety of pathological changes in cells, including mitochondrial dysfunction, DNA damage, telomere shortening, lipid peroxidation, and protein oxidative modification, all of which can trigger apoptosis and senescence. OS also induces a variety of aging-related diseases, such as retinal disease, neurodegenerative disease, osteoarthritis, cardiovascular diseases, cancer, ovarian disease, and prostate disease. In this review, we aim to introduce the multiple internal and external triggers that mediate ROS levels in rodents and humans as well as the relationship between OS, aging, and aging-related diseases. Finally, we present a statistical analysis of effective antioxidant measures currently being developed and applied in the field of aging research.

## 1. Introduction

The increasing global aging trend has brought the health status of the elderly and their associated diseases into sharp focus. Among many health concerns, aging and aging-related diseases are particularly prominent. Therefore, an in-depth investigation of the mechanisms of aging and their links to related diseases is essential for developing effective preventive and therapeutic strategies. Aging, an irreversible physiological process accompanied by the occurrence and development of organisms, is manifested by a gradual decline in the physiological functions of the body [1]. This process involves complex molecular and cellular changes, including mitochondrial dysfunction [2], DNA damage [3], telomere shortening [4], lipid peroxidation [5], protein oxidative modification [6], and other physiological activities. These changes not only accelerate the body’s aging process but also are important risk factors for various aging-related diseases.

As a crucial basic theory in the field of aging, the free radical theory of aging proposes that free radicals produced by external factors and internal metabolism can damage cell structure, block cell function, and ultimately trigger cell apoptosis and body aging. Oxidative stress (OS), as a physiological activity caused by free radicals, is a key factor leading to the oxidative damage of cells and tissues and is also the main driving force for aging and many diseases. A large number of studies on aging have shown that OS plays a central role in the onset and development of aging and related diseases. Persistent OS not only accelerates the aging process of the body but also further induces the occurrence of aging-related diseases by damaging important cell structures such as mitochondria, DNA, and telomeres. These diseases include retinal disease [7], neurodegenerative disease [8,9], osteoarthritis [10,11], cardiovascular diseases (CVDs) [12], cancer [13], and a variety of reproductive diseases [14,15], which seriously affect the quality of life and life expectancy of elderly individuals.

This article aimed to examine the relationship between OS and aging and the progress in aging-related disease research. Through an in-depth exploration of the mechanism of OS in aging and related diseases, we hoped to provide new ideas for the development and subsequent application of antioxidant strategies, help delay the aging process, prevent and treat aging-related diseases, and ultimately improve the quality of life and health of the elderly.

## 2. Triggers of OS

OS refers to an adverse status caused by the excessive generation of reactive oxygen species (ROS), which affects the normal physiological activities of cells [16,17]. OS increases with age and induces the development of various aging-related diseases. The triggers of OS are various and complex, and both external triggers (environmental factors, lifestyle, drugs, and toxic stimuli) and internal triggers (organelle dysfunction and increased NADPH enzyme activity) can mediate OS (Figure 1). Therefore, investigating the triggers of OS is essential for developing effective strategies to delay aging and prevent or treat various aging-related diseases.

### 2.1. External Triggers of OS

#### 2.1.1. Environmental Factors

Air pollution is a critical global concern. Research suggests that OS could be the main potential cause of respiratory system injuries due to air pollution. Numerous studies have investigated the relationships among atmospheric particulates (such as PM_2.5_ and PM_10_), gaseous pollutants (such as O_3_, SO_2_, NO_2_, and CO), and various OS biomarkers [18]. Li et al. [19] found that the exposure time of PM_2.5_ and SO_2_ was positively correlated with the level of OS biomarkers. Zeng et al. [18] demonstrated that exposure to PM_10_ is associated with elevated levels of inflammation biomarkers, such as interleukin (IL)-6, IL-8, tumor necrosis factor (TNF)-α, and 8-epi-pgf2α, which is closely related to OS. Other studies focusing on vulnerable groups (e.g., the elderly) have found that short-term exposure to NO_2_ and O_3_ and long-term exposure to CO were negatively correlated with fibrinogen levels [20]. However, some researchers have found no significant relationship between some atmospheric gaseous pollutants and OS biomarkers [21]. In summary, the complex particles and pollutants present in polluted air may be one of the causes of OS in the body.

In addition to air pollution, long-term exposure to ultraviolet (UV) radiation is also an important trigger of OS and may lead to skin aging, skin cancer, and other related diseases. Under normal physiological conditions, ROS produced by cellular respiration within mitochondria plays an important role in cell signaling and homeostasis regulation. However, under the environmental stress induced by excessive UV radiation, intracellular ROS levels significantly increase, leading to OS at the cellular or tissue level [22]. UV radiation is categorized into three types: UVA (320–400 nm), UVB (280–320 nm), and UVC (200–280 nm) [22]. Park et al. [23] demonstrated that UVB radiation induces skin aging and dehydration in mice, leading to OS and inflammation.

#### 2.1.2. Diet and Lifestyle

It has been reported that diets high in fat, sugar, and salt and low in fiber may be one of the triggers for OS. Diets rich in saturated and trans fatty acids increase the production of ROS, whereas antioxidant-rich foods such as fresh vegetables, fruits, and nuts help reduce OS. According to the study of the 20th Meeting of the Association of Academic European Urologists, several obese men who consumed healthier meals for 16 weeks, including a low-energy diet and a diet aligning with the National Health Service healthy eating recommendations based on the Eatwell Guidelines, showed reduced ROS levels, confirming that dietary habits are associated with ROS levels, which are associated with dietary habits [24].

In addition to unhealthy dietary habits, a lack of exercise and smoking habits may result in increased levels of ROS and enhanced OS. Studies have shown that exercise can increase the activity of antioxidant enzymes, reduce peroxide levels, improve antioxidant capacity, and protect cells from OS, thus preventing cell damage [25]. Amiri et al. [25] found that swimming reduced OS in male rats.

Smoking is a major cause of numerous non-communicable diseases (NCDs). Tobacco smoke contains a variety of toxic compounds, including high levels of ROS and reactive nitrogen species (RNS). Consequently, long-term exposure to tobacco smoke leads to severe OS and results in respiratory damage, which causes inflammation, chronic obstructive pulmonary disease, cancer, and other diseases [26]. Jansen et al. [27] demonstrated that smokers had elevated levels of OS biomarkers and imbalanced redox states compared to non-smokers in a population of healthy men, substantiating that smoking is a significant contributor to OS.

#### 2.1.3. Drugs and Toxic Stimuli

It has been suggested that OS is triggered by exposure to certain drugs or toxins, including heavy metals and industrial chemicals. For instance, methylphenidate (MPH), a stimulant of the central nervous system, is mainly used for the treatment of attention deficit and hyperactivity disorders [28]. Acute MPH administration has been reported to induce OS in the brains of young rats [29]. Numerous epidemiological studies have indicated that exposure to heavy metals, such as iron (Fe), mercury (Hg), manganese (Mn), copper (Cu), and lead (Pb), induces ROS production, reduces antioxidant levels in cells, and ultimately disrupts redox homeostasis. These processes can lead to OS, DNA damage, mitochondrial dysfunction, and apoptosis, potentially triggering a wide range of diseases, including neurodegenerative and CVDs [30,31].

### 2.2. Internal Triggers of OS

The excessive production of ROS is a major driver of cellular OS, leading to oxidative damage to proteins and DNA and playing a crucial role in the development of aging-related diseases. It has been reported that almost all intracellular compartments produce ROS as the byproducts of metabolic activities, and the mode of production depends on the cell type and the physiological state of the body [32].

The mitochondrial oxidative phosphorylation (OXPHOS) system is the center of cellular metabolism and a key pivot for energy production. At the cellular level, ATP is synthesized by mitochondria via OXPHOS, along with the generation of free radicals during electron transfer [33,34]. These free radicals derived from the mitochondria include ROS such as superoxide radicals (O_2_^•^), hydrogen peroxide (H_2_O_2_), and hydroxyl radicals (OH), as well as several RNS [35]. Complex I in the mitochondrial respiratory chain plays an important role in this context. The inhibition of complex I obstructs ATP synthesis and promotes ROS production, leading to respiratory failure and OS. Even a partial suppression of complex I increases mitochondrial ROS production, creating a “vicious circle” [36]. Numerous studies have shown that mitochondria-derived free radicals play a significant role in the process of cellular senescence and the development of aging-related diseases [35]. With the aging of the body, mitochondrial function deteriorates, which leads to increased ROS production and OS induction.

Several studies have shown that the endoplasmic reticulum (ER) regulates intracellular OS and that various physiological activities within the ER are important sources of ROS. ROS are mainly generated by the activity of cytochrome P450, NADPH oxidase (NOX) 4, and the oxidative protein folding processes mediated by endoplasmic oxidoreductin-1-like protein in the ER [32]. As the ER-resident proteins, protein disulfide isomerase and binding immunoglobulin protein have been reported to be oxidatively damaged and progressively dysfunctional during aging [37].

In addition to causing organelle dysfunction, NOX, which belongs to the NOX enzyme family, is another important intracellular source of ROS [38]. NOX mediates the production of ROS, transferring electrons from NADPH to O2 to produce superoxide anions (O2*−). This process is accompanied by the pumping of protons out of the cell membrane, creating a transmembrane potential difference that provides the driving force for protons to re-enter the cell and thereby contributing to cellular homeostasis. The transfer is regulated by flavin adenine dinucleotide, phosphorylated proteins, and Ca2+ concentration [39]. The factors that activate the NOX protein complex include insulin, growth factors, angiotensin, and TNF [37,39]. Under pathological conditions or when the organism is exposed to external stimuli, the activity of NADPH enzymes may be enhanced, leading to increased ROS production and OS.

## 3. Relationship between OS and Aging

Aging, a complex biological process, refers to the degradation, impairment, or even loss of tissue and organ function in the body over time [40]. In 1961, Hayflick et al. [41] discovered that cell proliferation is not unlimited and introduced the concept known as the “Hayflick limit”, which posits that cellular division is halted after a certain number of divisions are reached. Similarly, Harman et al. [42] put forward the “free radical theory of aging,” which suggested that OS could trigger a variety of aging-related damage, thereby promoting cellular aging. Several studies on aging have demonstrated that OS is one of the major drivers of cellular senescence and that excessive levels of ROS can hinder cell proliferation and accelerate cellular senescence. OS induces cellular senescence through various pathways, such as mitochondrial function, DNA damage, telomere length, lipid peroxidation, chronic inflammation, and oxidative modification of proteins (Figure 2).

### 3.1. OS Mediates Mitochondrial Dysfunction

Mitochondria are critical organelles in cells and are mainly involved in energy production and the regulation of signal transduction pathways. Under normal conditions, mitochondria produce ATP via the electron transfer process of the respiratory chain, and also produce a small amount of ROS as byproducts. However, when the level of ROS production exceeds the antioxidant capacity of cells, it causes OS and hinders physiological activities within the mitochondria.

OS can negatively affect mitochondria in multiple ways. Ye et al. [43] found that when male mice were exposed to arsenic, their ROS levels increased, which subsequently inhibited the NRF1/TFAM pathway via the SIRT1/PGC-1α axis. This led to impaired mitochondrial biogenesis and dynamics, ultimately leading to mitochondrial dysfunction. Wu et al. [44] showed that excessive ROS may disrupt the balance between mitochondrial fusion and fission, leading to aberrant changes in mitochondrial morphology. Such changes severely affect mitochondrial function and induce apoptosis. In addition to impaired dynamics and morphological abnormalities, it has been demonstrated that the mitochondrial membrane potential is significantly reduced with increased ROS, resulting in decreased ATP production and impaired energy metabolism, further activating apoptosis [45].

Several studies have indicated that OS may be triggered by mitochondrial dysfunction. For instance, abnormalities in the mitochondrial electron transport chain can result in the overproduction of ROS, which in turn triggers OS. Furthermore, mitochondria-related metabolic abnormalities may trigger OS [45]. In conclusion, there is a close relationship between OS and mitochondria. OS is capable of negatively affecting mitochondrial dynamics, morphology, and membrane potential, leading to mitochondrial dysfunction, which, in turn, may cause OS.

### 3.2. OS Mediates DNA Damage

DNA damage, one of the consequences of OS, is a permanent change in the DNA sequence during replication. The accumulation of DNA damage may lead to gene mutations, chromosomal aberrations, and other genomic instabilities, ultimately causing a wide range of diseases. ROS can react with the deoxyribose sugar, phosphate backbone, and nucleotide bases in DNA, leading to damages such as base modification, formation of DNA–protein crosslinks (DPCs), and the induction of single-strand breaks (SSBs) and double-strand breaks (DSBs) in the DNA molecule.

The oxidative modification of DNA bases has been reported to be one of the most common DNA damages, including SSBs or DSBs, DNA cross-linking, and changes in purine and pyrimidine bases and deoxyribose sugar [46]. Among these, DPCs can disrupt the normal structure of chromosomes, increase the instability of the genome, and are relatively difficult to repair. Zhang et al. [47] found that the concentration of DPCs gradually decreased in male mice with the increasing doses of antioxidant substances in the environment. This indicated that ROS levels are positively correlated with DPC concentrations.

Studies have shown that excessive ROS levels have certain direct or indirect negative effects on DNA molecules, such as attacking or generating hydroperoxides, leading to DNA SSBs or DSBs. Tolouee et al. [48] suggested that while excessive ROS levels lead to DNA strand breaks, the incomplete repair of DNA damage caused by persistently low ATP levels is the main trigger for DNA damage.

### 3.3. OS Mediates Telomere Length Reduction

As special structures at the ends of chromosomes, telomeres play a significant role in maintaining the stability of chromosomes and genomes. The shortening of telomeres as cells proliferate is considered an important marker of cellular senescence or apoptosis. Studies have demonstrated a close association between OS and telomere length [49].

Telomerase, a unique DNA polymerase, maintains telomere length by participating in the synthesis of telomeric DNA sequences. OS may accelerate telomere shortening by inhibiting telomerase activity. Assavanopakun et al. [50] found that OS conditions are commonly associated with reduced telomere length. Furthermore, telomerase activity is related to the duration of OS. Telomerase activity increases for a short period under OS; however, as the OS persists, telomerase activity gradually decreases, shifting its effect from potentially protective to detrimental. In addition, the repair of multiple damages to DNA caused by excessive ROS levels may lead to telomere shortening.

### 3.4. OS Mediates Lipid Peroxidation

Lipid peroxidation is the oxidative degradation of lipids in vivo in the presence of excessive ROS, resulting in the formation of lipid peroxides, such as malondialdehyde (MDA) and 4-hydroxy-2-nonenal, which lead to aging and related diseases [34,45,51]. 

Numerous studies have shown that OS is a major trigger for lipid peroxidation. Using red blood cells (RBCs) as the model, Celedón et al. [52] demonstrated the negative effect of OS on the lipid content in RBCs. Excess ROS react with lipid components within the cell membrane, resulting in significant damage to the morphology, structure, permeability, and overall function of the cell membrane. The lipid peroxidation reaction generates lipid peroxides that can cause a variety of DNA damage, such as the formation of DPCs and protein adducts [53].

Conversely, some studies have shown that lipid peroxidation also promotes OS. Irene et al. [54] found that lipid peroxidation, induced by OS, disrupts mitochondrial function, autophagy, and other normal physiological processes, thereby promoting lipid browning. The accumulation of lipid peroxides may increase the excessive accumulation of ROS, exacerbating OS and lipid peroxidation.

### 3.5. OS Mediates Protein Oxidative Modification

Proteins are important macromolecules in the body that are involved in most physiological activities. However, in the presence of functional abnormalities or as the body ages, proteins may suffer OS damage and undergo oxidative modifications, with the subsequent development of related diseases [55].

Most protein oxidative modifications have been reported to be induced either directly by ROS or indirectly through interactions with products of OS. As the state of OS continues, amino acid residues in proteins may be oxidized and the level of carbonyl compounds may rise, thereby activating the apoptotic program. Terao Ryo et al. [56] suggested that excess ROS inhibit the degradation of ubiquitinated proteins by downregulating the ROS proteasome, leading to the accumulation of protein oxidative modification products and accelerating apoptosis.

In addition, some scholars believe that the products of protein oxidative modification also exacerbate OS. For instance, Pérez-Ruiz et al. [57] found that certain amino acid residues, such as lysine, can exacerbate DNA damage by forming adducts with aldehydes produced by the peroxidation of polyunsaturated fatty acids. Overall, as with lipid peroxidation, there is a cyclical relationship between protein oxidative modifications and OS.

### 3.6. OS Mediates Exacerbation of Chronic Inflammation

Inflammatory aging is characterized by systemic chronic inflammation and is accompanied by cellular senescence, immune system aging, body aging, and aging-related diseases. Aging cells secrete senescence-associated secretory phenotypes, which promote chronic inflammation and induce immune system weakening, thereby resulting in persistently elevated levels of inflammation and ultimately leading to organ dysfunction and aging-related diseases [58]. Andrey et al. [59] demonstrated that the enhanced expression and activity of NOX4 during aging induces a proinflammatory phenotype in mouse vascular smooth muscle cells, exacerbating atherosclerosis.

Oxidative stress and inflammation are closely interrelated processes. Oxidative stress drives the NF-kB transcription factor pathway, which contributes to inflammatory cytokine production [60]. Garrido et al. [61] demonstrated that premature aging mice exhibited increased proinflammatory cytokine levels, reduced antioxidant defenses, and significant oxidative stress and inflammatory responses in adulthood, which are associated with premature immune system senescence and a shortened lifespan.

### 3.7. OS Induces Cellular Senescence through Other Pathways

In addition to mitochondrial function, DNA damage, telomere length, lipid peroxidation, and protein oxidative modification, OS induces cellular senescence through multiple pathways. For example, Jîtcă et al. [39] reported that OS can affect intracellular signaling pathways, leading to aberrant signal transduction. OS also accelerates organismal aging through the release of inflammatory cytokines [62], downregulation of immune function and regenerative capacity [63], induction of transcriptomic changes associated with the aging process [57], and inhibition of multiple oxidase activities [54]. Therefore, to delay cellular senescence and maintain normal physiological activities, it is necessary to adopt a multifaceted approach to counteract these effects by employing antioxidants, free radical scavengers, and DNA and protein damage repair methods.

## 4. OS and Aging-Related Diseases

With increasing age, the level of ROS rises, and dysregulated redox stress leads to a series of aging-related diseases, including retinal disease, neurodegeneration, osteoarthritis, CVDs, cancer, ovarian disease, and prostate disease. OS has now been shown to be a major factor in several aging-related diseases.

### 4.1. Retinal Disease

The retina is a nervous tissue formed by multiple cell layers that are responsible for the transmission and conversion of visual signals. With age, retinal function deteriorates, leading to a variety of retinal diseases, such as aging-related macular degeneration (AMD) and diabetic retinopathy (DR). Numerous studies have shown that OS is a major contributor to several retinal diseases.

Currently, OS is considered one of the main reasons for AMD. Abokyi et al. [64] found that the serum levels of several OS markers, including MDA and 8-hydroxy-2′-deoxyguanosine (8-OHdG), were elevated in patients with AMD, suggesting a close relationship between OS and AMD. The retinal pigment epithelium (RPE) is a major site vulnerable to OS damage, and the phagocytosis of photoreceptor outer segments is increased by prolonged exposure to exogenous triggers such as visible light and UV radiation. This degradation of outer segments leads to lipofuscin accumulation, resulting in RPE cell dysfunction [65]. Abnormal RPE cells can increase ROS levels, forming a vicious cycle and further aggravating OS. In addition, OS blocks mitochondrial function, thereby inducing RPE dysfunction and AMD development [66].

DR is an ophthalmic complication of diabetes mellitus (DM) and is generally defined as a type of microangiopathy [65]. Long-term hyperglycemia in patients with diabetes leads to increased ROS production, making the microvessels more vulnerable to oxidative stress. The disturbance of redox homeostasis leads to retinal neuronal death, a subsequent breakdown of the blood retinal barrier and increased vascular permeability, leading to DR [67]. In addition, some molecules are altered and pathways are adversely activated, such as the formation of advanced glycation end products, the polyol pathway, the protein kinase C pathway, and the hexosamine pathway [68]. These pathways are both activated by and contribute to excess ROS levels in the body. Excessive cytosolic ROS interact negatively with mitochondria, creating a “vicious cycle” that activates apoptosis in retinal capillaries. Multiple factors combine to increase OS and ultimately lead to the development of DR. Studies have shown that the accumulation of ROS blocks the normal physiological activity of the mitochondria, and that mitochondrial dysfunction affects the production of ROS in retinal cells, activity of optic nerve cells, and function of photoreceptors. Moreover, mitochondrial dysfunction may reduce mitochondrial energy production, leading to optic nerve degeneration [67].

### 4.2. Neurodegeneration

Neurodegenerative disease encompasses a variety of disorders that affect the central nervous system, including Parkinson’s disease (PD) and Alzheimer’s disease (AD). These conditions are characterized by progressive neuronal degeneration or death of neurons, resulting in a gradual decline in everyday functions such as cognition and movement [69]. OS has been reported to be a key mechanism in the development of neurodegenerative disease [7].

PD is a neurodegenerative disorder that affects the motor system. It is characterized by the death of nigrostriatal dopaminergic neurons, and OS is a recognized cause of the triggering mechanism. Morales-Martínez et al. [70] used α-synuclein transgenic rats and found that the levels of several biomarkers of OS were positively correlated with nerve degeneration in several regions of the rat brains. As ROS levels increase, lipid peroxidation (LP) and mitochondrial dysfunction occur, triggering neuronal death, and the decrease in neuronal numbers promotes ROS production. The “vicious cycle” of neuron-ROS occurs constantly and eventually activates PD. In addition, PPARα has been shown to be associated with the onset of PD. The deficiency of PPARα disrupts complex I of the mitochondrial respiratory chain, induces fatty acid oxidation, aggravates OS, and triggers PD.

OS is a major trigger factor in the development of AD, similar to its role in PD. As a major phenotype of AD, β-amyloid (Aβ) deposition triggers OS, leading to neuronal damage and even cell death. It has been shown that the ROS scavenger N-acetylcysteine attenuates Aβ deposition through the activation of the PI3K/Akt/GLUT1 pathway while improving impaired glucose uptake [71].

### 4.3. Osteoarthritis

With increasing age, the cartilage and bone undergo degenerative changes, leading to degenerative joint diseases, such as osteoarthritis. Altered redox homeostasis is the cause of multiple degenerative joint diseases.

Due to the imbalance of decomposed or synthetic signals in chondrocytes, the extracellular matrix is progressively broken down, which is one of the main phenotypes of osteoarthritis [17]. Collins et al. [72] found that peroxiredoxin, an important intracellular antioxidant, was over-oxidized in cartilage sections and osteoarthritic cartilage from older adults, which was prevented by cartilage explants targeting mitochondrial catalase (MCAT) and MCAT transgenic mice. This demonstrates that aging-related OS can disrupt normal physiological signaling in the body and lead to osteoarthritis.

### 4.4. CVDs

CVDs, including heart disease and atherosclerosis, are a major health concern worldwide. These diseases are associated with a variety of NCDs, including hypertension and hyperglycemia, which are often exacerbated by long-term unhealthy habits such as a high-sugar diet, smoking, and excessive alcohol consumption. Numerous studies have shown that OS plays an important role in the pathogenesis of CVD.

On the one hand, at normal concentrations, ROS help maintain the redox balance of the body, contribute to the transcription of normal cells, protect the function of the cardiovascular system, and play an anti-atherosclerotic role and other beneficial effects [73]. On the other hand, with the aging of the body, ROS levels rise and endothelium-derived contracting factors are stimulated, leading to endothelial dysfunction and CVD induction. Interactions between some free radicals, such as the superoxide anion (O_2_^•−^) reacting with NO to form ONOO^−^, promote endothelial cell dysfunction and even lead to apoptosis [73].

It has been shown that the decreased activity of the transcription factor Nrf2 also increases ROS levels and exacerbates vascular dysfunction caused by metabolic diseases during aging. Nrf2 is mainly responsible for improving antioxidant defenses, hindering adverse reactions such as inflammation, and maintaining the balance of normal physiological activities. Under normal physiological conditions, Nrf2 is hindered by the negative regulator Keap1, leading to its ubiquitination [74]. However, when the concentration of ROS is excessively high, KEAP1 activity is decreased or even inhibited, leading to decreased Nrf2 degradation. Nrf2 then accumulates in the nucleus. With aging, Nrf2 activity decreases, susceptibility to OS increases, and the cell sustains a state of excess ROS, triggering CVD [55].

### 4.5. Cancer

Cancer is a complex disease that involves the interaction of multiple factors. High ROS levels are an important distinction between cancer cells and normal cells. Studies have shown that the promoting effect of OS on cancer is related to the multiple DNA site damages it causes, including modifications to nucleobases, the formation of DPCs, and DNA strand breaks. 8-OHdG has been reported to be elevated in various types of cancer and is a well-established biomarker of OS [75].

Meanwhile, OS can also harm cancer cells. To repair DNA damage caused by OS, the activity of DNA repair enzymes increases, thereby depleting the ATP reserves needed for cancer cell survival. Furthermore, if the state of OS persists, the increased levels of ROS in cancer cells could activate apoptosis [35].

### 4.6. Other Diseases

In addition to its role in retinal disease, neurodegenerative disease, osteoarthritis, CVDs, and cancer, OS can also trigger a variety of reproductive diseases, such as ovarian diseases and prostatic diseases. The “free radical theory” proposed by Harman et al. [42] suggests that OS caused by excess ROS is the most important cause of mammalian cellular senescence. Numerous reproduction-related studies have shown that OS induces a variety of injuries, including the shortening of telomeres, mitochondrial dysfunction, and the initiation of inflammatory responses, leading to ovarian diseases [15,76].Excess ROS also lead to decreased ovum quality, delayed ovum maturation, blocked signaling between oocytes and germ cells, and severely affected ovulation [76].

Additionally, studies have demonstrated that patients with aging-related lower urinary tract disorders (LUTS), which are a common prostatic disease, exhibit elevated levels of OS biomarkers [16]. There is a close relationship between mitochondrial dysfunction due to OS and the development of LUTS. It has also been reported that CVD may increase the risk of prostatic diseases through the pathway of OS [76].

## 5. Antioxidant Strategies

Whenever aerobic respiration and cell metabolism occur in the body, redox reactions occur and ROS are produced. When the ROS concentrations are at normal levels, a balance between ROS and the body’s antioxidant system ensures cellular homeostasis. However, when ROS are overproduced and antioxidant enzymes are insufficient, OS occurs, leading to apoptosis and tissue damage [77].

To repair the oxidative damage caused by ROS, various artificial interventions with antioxidants have been employed. At present, antioxidants are broadly categorized into synthetic and natural compounds [78]. Synthetic antioxidants include butyl hydroxyanisole, butyl hydroxytoluene, and propyl gallate. Natural antioxidants include flavonoids (e.g., proanthocyanidins and licorice extract), polyphenols (e.g., tea polyphenols and resveratrol), vitamins and their derivatives (e.g., vitamin E, vitamin C, and coenzyme Q10), antioxidant peptides (e.g., soybean peptides and glutathione), active polysaccharides, and minerals in trace amounts (e.g., selenium, silicon, and zinc). 

Different antioxidants have different mechanisms of action because of their different active groups. Synthetic antioxidants neutralize free radicals by donating an H atom (hydrogen atom transfer) or via a single electron transfer mechanism [79]. Phenolic compounds induce the expression of genes encoding antioxidant enzymes and phase II detoxification enzymes by stimulating Nrf2 to enter the nucleus and bind to the antioxidant response element after Keap1 uncoupling, and promote the production of antioxidant enzymes and phase II detoxification enzymes [80]. Coenzyme Q10 is a mitochondria-targeted antioxidant that protects stem cells from OS-induced senescence by affecting the Akt/mTOR signaling pathway, thereby maintaining their proliferation balance [81]. SkQ1 exerts its antioxidant properties through fatty acid-mediated uncoupling, neutralization of lipid peroxide radicals, and regulation of electron flow at the mitochondrial level [82]. Astaxanthin can induce the antioxidant enzyme paroxoanase-1, increase glutathione concentration, and prevent lipid peroxidation in cultured hepatocytes [83]. Most human selenoproteins are oxidoreductases containing the amino acid selenocysteine (SeCys) at their catalytic sites. In the antioxidant enzyme GPX, SeCys residues use glutathione as a substrate to catalyze the reduction in hydrogen peroxide and hydrogen peroxide radicals, thereby reducing free radicals and DNA damage [84,85].

The emergence of various types of antioxidants has expanded the range of options available for combating OS. However, research reports [86] indicate that synthetic antioxidants may pose risks such as potential carcinogenesis, cytotoxicity, and endocrine disruption with excessive or improper use [87]. Synthetic oxidants also exhibit drawbacks related to high-temperature resistance and stability [88]. By contrast, natural antioxidants, which are mainly derived from vegetables and fruits, are considered safer, but their low bioavailability leads to poor therapeutic efficacy. At present, certain drug delivery strategies have been shown to improve drug bioavailability and enhance their therapeutic effects. Nanomaterials are widely used in drug delivery systems as they can be designed and functionalized according to the nature and needs of natural antioxidants and offer an efficient and safe in vitro delivery option with excellent application value and development prospects [89].

In addition to the artificial intervention of antioxidants, daily lifestyle habits play an important role in the aging process of the body. Adopting a healthy lifestyle, including a balanced diet, proper physical exercise, and optimism, can significantly improve the body’s antioxidant capacity, delay aging, and prevent diseases.

## 6. Conclusions and Future Directions

OS is an important factor in the aging process and the development of various aging-related diseases. As humans age, under the influence of various internal and external factors, the excessive production of ROS in the body triggers various biochemical reactions, leading to cellular aging and the occurrence of various related diseases. Based on current research on OS and aging, we summarized six internal/external triggers of OS. They induce ROS production through different mechanisms, thereby activating OS responses and leading to pathological changes such as mitochondrial dysfunction, DNA damage, telomere shortening, lipid peroxidation, and oxidative modification of proteins. In addition, we further analyzed the relationship between OS and major aging-related diseases such as the significant effects of OS on retinal disease, neurodegenerative disease, osteoarthritis, CVDs, cancer, and some reproductive diseases. Antioxidants have attracted much attention owing to their potential therapeutic value, as they can intervene in different molecular mechanisms of OS, providing a new perspective for the prevention and treatment of aging-related diseases. This article introduced the main antioxidants discovered so far, aiming to find therapeutic drug targets for the different molecular mechanisms of OS-induced diseases in the body and use more effective and safer interventions against aging. A deep understanding of the relationship between OS and aging-related diseases not only helps to develop new treatment methods but also provides a theoretical basis for the formulation of personalized treatment strategies. A comprehensive understanding of this field will open up new avenues for future medical research and practice, helping us better understand and intervene in the aging process and related diseases.

## Figures and Tables

**Figure 1 antioxidants-13-00394-f001:**
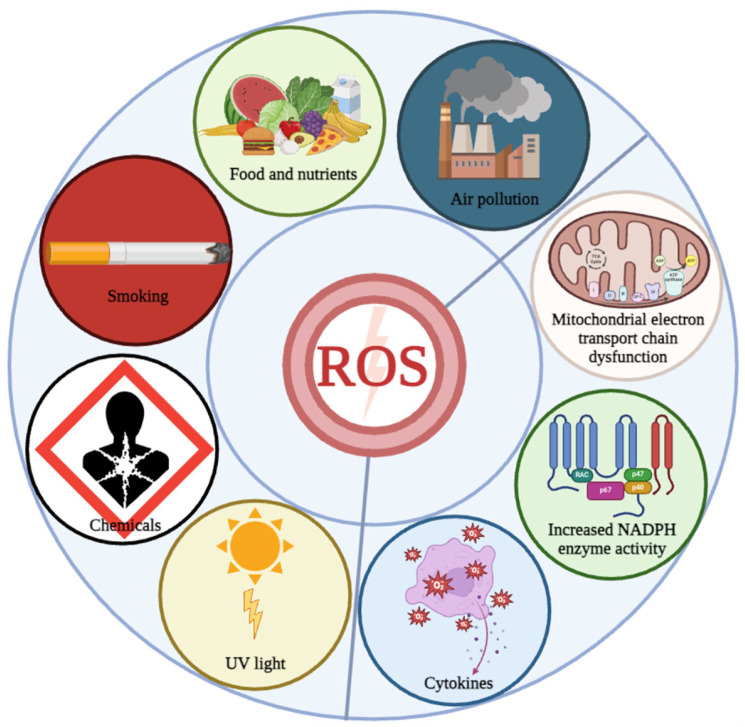
Triggers of OS. Oxidative stress is induced by internal and external triggers. Internal triggers include the mitochondrial electron transport chain dysfunction, increased NADPH activity, etc., while external triggers include air pollution, food and nutrients, smoking, chemicals and UV light.

**Figure 2 antioxidants-13-00394-f002:**
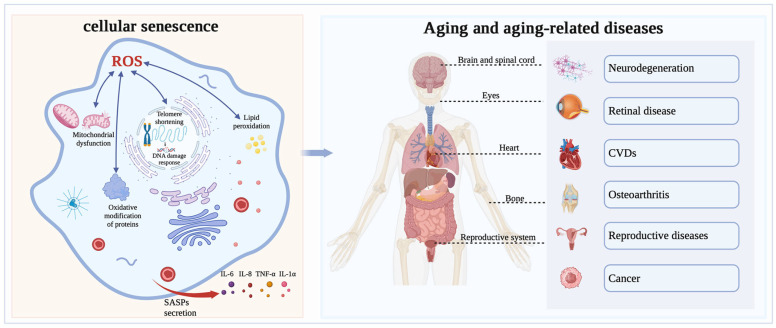
The relationship between OS and aging. Intracellularly, ROS lead to mitochondrial dysfunction caused by mitochondrial enlargement and rupture, leading to a reduction, breakage and ribosome detachment in the rough ER, resulting in DNA damage, shortening of telomeres, lipid peroxidation and oxidative modification of proteins, and triggering cellular oxidative stress and senescence. Notably, cellular senescence secretes senescence-associated secretory phenotypes (SASPs), including IL-6, IL-8, and TNF-α, which promote chronic inflammation and ultimately lead to systemic aging. Oxidative stress-triggered aging can cause a variety of diseases, such as retinal disease, neurodegenerative disease, reproductive diseases, osteoarthritis, cardiovascular diseases and other diseases.

## Data Availability

No data were used for the research described in this article.

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
