# Peer review of "Progress in Understanding Oxidative Stress, Aging, and Aging-Related Diseases"

_antioxidants, 2024, doi:10.3390/antiox13040394_

Round 1

Reviewer 1 Report

Reviewer’s comments

Manuscript title: Advances in oxidative stress, aging, and age-related diseases.

Authors: Yumeng Li, Juyue Luo, Xutong Tian, Yaping Zhao, Jianying Yang and Xin Wu 

This manuscript by Li Y. et al. reviewed the origin of oxidative stress and the relationship of oxidative stress to aging and age-related disease. The authors comprehensively covered the most important diseases/conditions that may be caused by excessive oxidative stress. The manuscript also covered a list of the most important anti-oxidant strategies. Finally, a brief conclusion was put to end the manuscript.  

While the manuscript is comprehensive, the contents are in lack of novelty. It therefore fits well as an instructional review, if such is the editorial purpose. 

The writing demands a major revision. There are a lot of points in need to be taken care of.

The writing demands a major revision. There are a lot of points in need to be taken care of. Just list a few examples here:

1.      Lines 11- 12: may be revised as ‘are produced through redox reactions as byproducts of respiratory and metabolic activities.’

2.      Line 16- 18: ‘a variety of age-related diseases or conditions such as retinal aging, neurodegeneration, joint aging, cardiovascular diseases, cancer, ovarian aging, and prostate diseases.’ It is important to distinguish normal aging (or even healthy aging) from diseases. Aging itself is not a disease. The same situation applies to other contexts in this manuscript (for example lines 291-293).

3.      Line 297: ‘……conversion and transmission of visual signals…’ would better be revised as‘……transmission and conversion of visual signals…’ .

4.      Line 299: ‘Numerous studies have shown that OS is a major contributor to several retinal diseases.’

5.      Line 306-307: This degradation of outer segments leads to lipofuscin accumulation, resulting in RPE cell dysfunction.

6.      Line 312: as a type of microangiopathy……

7.      Lines 313- 315: Some molecules are altered and pathways are adversely activated, such as the formation of advanced glycation end products, polyol pathway, protein kinase C pathway, and the hexosamine pathway.

8.      In Line 263: ROS inhibits…., but in Line 316-317, cytosolic ROS interact.

ROS would better be used as a plural noun throughout the entire manuscript.

9.      Lines 50- 51: …..and the progress in age-related disease research.

10.  Lines 52-53: for the development and subsequent application of antioxidant strategies …….

11.  Line 57: OS refers to an adverse status caused by excessive generation of reactive oxygen…..

Disorders have been on a disease status.

12.  Lines 63-64: Therefore, investigating the triggers of OS is essential to develop effective strategies to delay aging and prevent or treat various aging-related diseases.

13.  Line 66: Figure 1. Triggers of OS. Oxidative stress is induced by internal and external triggers.

14.  Line 94: 2.1.2. Diet and lifestyle

15.  Line 98: AAEU ? study, give the full name when an abbreviation first appears.

16.  Line 172: can hinder cell proliferation…. Remove the specification on this word.

17.  Lines 201-202: is a permanent change in DNA sequence during replication

18.  Lines 239-240: ‘The relationship between OS and lipid peroxidation has been reported.’ This entire sentence has to be deleted.

19.  Line 248: Reversely, some studies have shown that lipid peroxidation also promotes OS.

20.  Line 250: The accumulation of lipid peroxides may further……..

21.  Line 256-257: proteins may suffer OS damage and undergo oxidative modifications, with subsequent development of related diseases.

22.  Line 279: …….and maintain normal physiological activities, it is necessary……

23.  Line 299: Numerous studies have shown that OS is a major contributor to several retinal diseases

24.  Line 331: As ROS levels increase, LP ? and mitochondrial dysfunction…(give the full name when an abbreviation first appears).

25.  Line 344-345: It has been established that altered redox homeostasis is the cause of multiple degenerative joint diseases… (when it has already been established, it cannot be just a potential cause)

26.  Line 350: MCAT transgenic group (give the full name when an abbreviation first appears).

Author Response

In accordance with your suggestion, we have revised the language of the article, and invited professionals to the article for the full language Polish, the proof of the revision attached to the reply. The details of the changes have been marked in red in the article.

Reviewer 2 Report

Include a section on the difference between aging and senescence as you mention the term senescence-associated secretory phenotype (SASP) in figure 2.

Not other comments except those mentioned above

Author Response

 As you suggested, we have added the difference between aging and senescence “Notably, cellular senescence secretes senescence-associated secretory phenotypes (IL-6, IL-8, TNF-α, etc.) that promote chronic inflammation and ultimately lead to body aging” in lines 331-333, and highlighted in red.

Reviewer 3 Report

This review paper by dr. Li and colleagues examines the sources of oxidative stress for the organism and the mechanisms by which oxidative stress influences aging and is involved in age-related diseases. The text is very well organized and well written. None of the specific topics (for instance the OS-related pathologies) is discussed in detail, but this is in favor of a comprehensive description of the main topic.

Figures: The figures should be cited in the text.

Line 78-79: IL6, IL8, and TNFalpha are usually employed as inflammatory markers, and not as OS biomarkers. In addition, it is known that OS and inflammation may induce / influence each other in a reciprocal fashion. Inflammation is also a condition that, unfortunately, becomes usual as the age advances, therefore a section analyzing the relationships between OS and inflammation should be included in this paper.

Line 229: the exact meaning of “statistically concluded” is not totally clear to me.

Lilne 296: I don’t think the expression “sensory tissue” is correct. The different tissues are defined on the basis of their histologic characteristics (e.g. muscular tissue, …), and “sensory” is not such a type of feature. The retina is made of nervous tissue.

Line 311-319: DR is certainly characterized by OS, but the effect of OS is not (or at least not only) the death of retinal endothelial cells. Indeed, major neuronal cell death has been reported in DR and this disease has been proposed as a neurodegenerative (more than a microvascular) disease of the retina. Please check the literature to better consider the role of OS in DR.

Line 321-322: Relative to the previous point: the retina is also part of the central nervous system, therefore it is logic that neurodegeneration is a major effect of OS, as in other regions of the central nervous system.

Table 1: The natural substances that have been described to play antioxidant effects in different organs or tissues sum up to a very large number and they cannot be represented in a simple table. The compounds listed here (and the relative bibliographic references) are only a very minor part of the whole and it is not clear why these compounds are considered the “main antioxidants”. I suggest deleting Table 1.

Line 437- 440: The low effectiveness of natural antioxidants is mainly due to their low bioavailability and current research is aimed at improving bioavailability through, for instance, nanocarrier-mediated delivery strategies. The Authors may spend a couple of lines to account for this promising perspective.

References: It seems that most (if not all) references in the reference list report the full first name and the initial of the surname of the authors, which is quite uncommon.

Author Response

Thank you very much for your suggestions, we have changed each item in the article, where the changes have been marked in red.

Figures: The figures should be cited in the text.

Reply: Thank you so much for your careful check. We have cited the figures in the text.

Line 78-79: IL6, IL8, and TNF alpha are usually employed as inflammatory markers, and not as OS biomarkers. In addition, it is known that OS and inflammation may induce / influence each other in a reciprocal fashion. Inflammation is also a condition that, unfortunately, becomes usual as the age advances, therefore a section analyzing the relationships between OS and inflammation should be included in this paper.

Reply: As you suggested that we have added OS-mediated exacerbation of chronic inflammation to the section on the relationship between OS and aging in lines 300-314, and highlighted in red.

Line 229: the exact meaning of “statistically concluded” is not totally clear to me.

Reply: Thank you for pointing out this problem in manuscript. We have modified “statistically concluded” to “found” in line 261, and highlighted in red.

Line 296: I don’t think the expression “sensory tissue” is correct. The different tissues are defined on the basis of their histologic characteristics (e.g. muscular tissue, …), and “sensory” is not such a type of feature. The retina is made of nervous tissue.

Reply: Thank you so much for your careful check. We have replaced the sensory tissue with nervous tissue in line 342, and highlighted in red.

Line 311-319: DR is certainly characterized by OS, but the effect of OS is not (or at least not only) the death of retinal endothelial cells. Indeed, major neuronal cell death has   been reported in DR and this disease has been proposed as a neurodegenerative (more than a microvascular) disease of the retina. Please check the literature to better consider the role of OS in DR.

Line 321-322: Relative to the previous point: the retina is also part of the central nervous system, therefore it is logic that neurodegeneration is a major effect of OS, as in other regions of the central nervous system.

Table 1: The natural substances that have been described to play antioxidant effects in different organs or tissues sum up to a very large number and they cannot be represented in a simple table. The compounds listed here (and the relative bibliographic references) are only a very minor part of the whole and it is not clear why these compounds are considered the “main antioxidants”. I suggest deleting Table 1.

Reply: Considering the Reviewer’s suggestion, we have deleted Table 1 and the relevant citations.

Line 437- 440: The low effectiveness of natural antioxidants is mainly due to their low bioavailability and current research is aimed at improving bioavailability through, for instance, nanocarrier-mediated delivery strategies. The Authors may spend a couple of lines to account for this promising perspective.

Reply: Thank you for the above suggestions. We have described in lines 490-497 that drug delivery systems can be an important strategy to increase the bioavailability of natural antioxidants, elucidating their wide range of applications and prospects for development.

References: It seems that most (if not all) references in the reference list report the full first name and the initial of the surname of the authors, which is quite uncommon.

Reply: We have modified the format of the ref.

The full text of the English please professional personnel to modify, the polish certificate attached at the end of the article.

Round 2

Reviewer 1 Report

All the points have been responded and revised accordingly in the manuscript by the authors.

All the points have been responded and revised by the authors.

Reviewer 2 Report

This manuscript has been improved and to my point of view can be published

nothing

Reviewer 3 Report

The Authors have significantly improved the manuscript and I have no further comments. However, the references should be checked again, since: i) in several places in the text the advice “Error! Reference source not found” is displayed; ii) in the reference list there are still references in which the authors’ names are in full while only the initials of the surnames are indicated.

Please see comments above.
